# A Strategy for Minimizing Circulatory Arrest Duration in Complex Aortic Arch Procedures

**DOI:** 10.3390/medicina59061007

**Published:** 2023-05-24

**Authors:** Robert Balan, Petar Soso, Parwis Massoudy, Till Proschek, Wiebke Kurre, Christian Mogilansky

**Affiliations:** 1Department of Cardiac Surgery, Klinikum Passau, 94032 Passau, Germany; petar.soso88@gmail.com (P.S.); parwis.massoudy@klinikum-passau.de (P.M.); christian.mogilansky@klinikum-passau.de (C.M.); 2Department of Vacular Surgery, Klinikum Passau, 94032 Passau, Germany; till.proschek@klinikum-passau.de; 3Department of Radiology, Klinikum Passau, 94032 Passau, Germany; wiebke.kurre@klinikum-passau.de

**Keywords:** aortic aneurysm, aortic dissection, frozen elephant trunk

## Abstract

*Background and Objectives*: Aortic arch pathologies represent a surgical challenge. The challenge is partly due to the necessity of complex cerebral, visceral, and myocardial protection measures. Aortic arch surgery generally requires a significant duration of circulatory arrest, which includes deep hypothermia levels with the associated sequelae. This retrospective observational study shows the feasibility of a strategy that reduces circulatory arrest duration and eliminates the need for deep hypothermia during the procedure. *Materials and Methods*: Between January 2022 and January 2023, 15 patients (n = 15) with type A aortic dissection underwent total arch replacement with a frozen elephant trunk. Cardiopulmonary bypass and organ perfusion were established via arterial lines in the right axillary artery and one of the femoral arteries. In the latter vessels, a y-branched arterial cannula was used (ThruPort^TM^_)_, allowing for endo-clamping of the stent part of the frozen elephant trunk with a balloon and subsequent perfusion of the lower body. *Results*: Applying this modified perfusion technique, circulatory arrest time could be reduced to a mean of 8.1 ± 4.2 min, and surgery was performed at a mean lowest body temperature of 28.9 ± 2.3 °C. The mean ICU and hospital stays were 18.3 ± 13.7 days and 23.8 ± 11.7 days, respectively. The rate for 30-day survival was 100%. *Conclusions*: By applying our modified perfusion technique, the circulatory arrest duration was below ten minutes. As a consequence, deep hypothermia could be avoided, and surgery could be performed at moderate hypothermia. Future studies will have to show whether these changes may be translated into a clinical benefit for our patients.

## 1. Introduction

Open surgical repair of the aortic arch, especially when complicated by dissection, remains one of the most complex procedures in adult cardiac surgery. The operation is long lasting, technically demanding and often combined with considerable ischemia for the heart and lower body.

During the last 50 years, surgery of the aortic arch has been usually performed in deep hypothermia and circulatory arrest, thus under time pressure, making the surgery stressful and exhausting.

Meanwhile, selective brain perfusion has become standard in aortic arch surgery. Simultaneous perfusion of the brain and of the heart has been described [1] and has been applied in some centers. However, the lower body is generally not perfused for 30–60 min, requiring circulatory arrest in deep hypothermia (22–26 °C).

Herein, we describe a method to reduce circulatory arrest duration of the lower body by beginning perfusion as soon as the frozen elephant trunk is in place, even before the distal suture line is completed.

The strategy further increases the safety of the procedure by significantly reducing the circulatory arrest duration, enabling the surgeon to perform the aortic arch procedures at moderate hypothermia preventing the sequelae associated with deep hypothermic circulatory arrest.

## 2. Materials and Methods

### 2.1. Study Design and Objective

We conducted a retrospective, observational study, evaluating patients who underwent emergent surgical treatment of a type A aortic dissection with a novel perfusion technique avoiding deep hypothermic circulatory arrest.

Between January 2022 and January 2023, fifteen (n = 15) consecutive, non-selected patients underwent surgical correction for type A acute aortic dissection at our institution. A frozen elephant trunk was implanted in all 15 patients. Five patients received concomitant/hybrid intervention, three of them before surgery and two of them during or immediately after the procedure. Preoperative diagnostic procedures were performed according to a standardized dissection protocol as described later. The aim of the present investigation was to evaluate the results of our modified perfusion strategy.

### 2.2. Preoperative Evaluation

When a patient with aortic dissection was admitted to our hospital, a multidisciplinary team consisting of a radiologist, a vascular surgeon, an anesthetist, and a cardiac surgeon was always involved.

A standardized, comprehensive ECG-triggered, dual phase CT scan of the aorta and major arterial branches, extending from the circle of Willis to the groin, was performed in each patient. Special focus was placed on the patency of the circle of Willis and the detection of any anatomical abnormalities in order to plan cannulation and cardiopulmonary bypass (CPB) strategies. In the majority of cases, dual peripheral arterial cannulation was implemented via one femoral artery and the right axillary artery; additional perfusion via the left subclavian artery or the left carotid artery may have been necessary in cases with an incomplete circle of Willis and/or a lacking junction of the vertebral arteries.

According to the extent of the disease, the presence of malperfusion, and the clinical status of the patient, an intervention, aimed at the restitution of adequate perfusion of the brain or visceral or peripheral arteries, may have been performed before or during surgery.

The following imaging parameters were assessed to guide further surgical and endovascular treatment:The presence and size of the major intracranial collaterals (the anterior communicating artery, the posterior communicating arteries, and the junction of the vertebral arteries).The extension of the dissection including the involvement of supra-aortic and visceral branches and peripheral arteries.The size and involvement of the groin vessels.The size of each supra-aortic main vessel (the brachiocephalic trunk, the left carotid artery, and the left subclavian artery).The size of the aorta in zones 0, 1, 2, and 3.

### 2.3. Anesthesia

Standard cardiac anesthesia techniques (i.v. sufentanil 0.5 μg/kg/h, etomidate 0.25 mg/kg, pancuronium 0.1 mg/kg, sevoflurane, and propofol 3 mg/kg/h) were used for the induction and maintenance of anesthesia. Invasive monitoring was performed with dual arterial pressure lines (placed usually in the right radial artery and one of the femoral arteries) as well as one central venous catheter. A single-lumen tube was used for ventilation. All patients had intraoperative transesophageal echocardiography to help position guidewires for femoral/axillary arterial and venous cannulation as well as to monitor cardiac function. Near-infrared spectroscopy sensors were placed on the right and left forehead and, in increasing numbers, directly adjacent to the right and left mastoid bone.

### 2.4. Technique

The surgical procedure was performed in a hybrid operating room, facilitating diagnostic and interventional options for radiologists and vascular surgeons. Eventual malperfusion syndromes were addressed at the beginning of the surgical procedure.

The primary arterial cannulation site was the right axillary artery via open access. A second, y-branched arterial cannula (ThruPort^TM^, Edwards LifeSciences, Irvine, CA, USA) was placed either percutaneously, under ultrasound guidance, or, alternatively, with open access, into the femoral artery. Under fluoroscopic and TEE guidance, a guide wire was advanced through the side port of the y-branched femoral arterial cannula into the true lumen of the descending aorta, and a non-insufflated occlusion balloon was placed at the diaphragm level of the aorta by the radiologist. Venous cannulation may have been performed percutaneously via the right femoral vein or through the right atrium after sternotomy. After dual arterial cannulation and placement of the deflated occlusion balloon, CPB was started, aiming at moderate hypothermia (28–30 °C). If necessary and based on the radiological assessment of the intracranial collaterals, the left axillary artery or the left carotid artery was additionally cannulated to maintain optimal cerebral perfusion.

Surgery was continued with the replacement of the proximal aorta (the heart-first technique) including aortic root surgery (either replacement or reconstruction). Reconstruction included stabilization of the sinutubular junction and application of glue into the inter-layer space in the non-coronary sinus in case it was the only sinus affected by dissection. The non-coronary sinus showed signs of dissection in almost every case included in this study. With the aortic root and ascending aorta completed, we started normothermic heart perfusion with 350–500 mL/min at a perfusion pressure of 80–110 mmHg via a further arterial line (usually the cardioplegic line), which was inserted into the aortic ascending graft. At this point of the operation, the patient was under a triple perfusion regimen separated into the upper body, including the brain, the lower body, and the heart.

Before the transition from total body perfusion to isolated brain perfusion, the effectiveness of unilateral cerebral perfusion was examined by temporary occlusion of the brachiocephalic trunk and the left common carotid artery. In case of a critical decrease in NIRS values on the left side (the forehead or mastoid bone), an additional perfusion catheter was placed into the left common carotid artery (via direct access). The supra-aortic vessels were then ligated close to their origin from the aorta. Thereafter, uni- or bilateral cerebral perfusion was started. Meanwhile, a core temperature of 28 °C to 30 °C had been reached and hypothermic circulatory arrest of the lower body was commenced. The aortic arch was now transected in zone 0. The introduction of the hybrid prosthesis into the descending aorta was controlled by the guide wire, previously placed into the true lumen over one femoral artery. After deployment of the stent, the femoral occlusion balloon was advanced into the stented part of the hybrid prosthesis and insufflated. Perfusion of the lower body was resumed via the femoral arterial line (Figure 1). The duration of circulatory arrest performed at moderate hypothermia was less than ten minutes. Selective perfusion of the lower body, the brain, and the heart was now being performed simultaneously. Since the advent of the EVITA^TM^ open Neo stent-prosthesis (CryoLife, Kennesaw, GA, USA), the trifurcated version of the prosthesis has been used in the majority of cases.

Having performed the distal anastomosis in zone 0 and the connection between the stent-graft prosthesis and the prosthesis for the ascending aorta, the system was de-aired, the balloon was deflated, and isolated heart perfusion was terminated. Body and heart perfusion was now accomplished via the femoral arterial line. Brain perfusion was continued via the axillary line(s).

Once the continuity of the aorta was restored, the left carotid artery, the brachiocephalic trunk, and the left subclavian artery were trimmed, and end-to-end anastomoses with the trifurcated branches of the vascular graft (EVITA open Neo^TM^) were performed in extra-anatomic fashion. Thereafter, whole-body perfusion was continued via the line in the right axillary artery until the end of CPB. In selected cases, the left subclavian artery may have stayed ligated and perfusion via the left subclavian artery may have been sacrificed if the intracranial collaterals were sufficient and the NIRS values stayed unremarkable during surgery.

Immediately after surgery, we performed a CT scan of the aorta and major arterial branches from the circle of Willis to the groin in order to guarantee adequate whole-body perfusion or to indicate further interventional procedures.

## 3. Results

In the time interval between January 2022 and January 2023, 15 patients were operated on using the perfusion technique as described above. The pre-operative patient characteristics are described in Table 1.

A comprehensive overview of our intra-operative patient characteristics is shown in Table 2. Of note, perioperative carotid intervention was necessary for 5 patients (33%), the lowest body temperature was 28.9 °C, and the mean circulatory arrest duration (at moderate hypothermia) was 8.1 min.

The postoperative patient characteristics are summarized in Table 3. The 30-day survival was 100%. Seven patients had neurological deficits postoperatively: five of these patients had one-sided hemiplegia and two of them had one-sided arm weakness. It is worth mentioning that three of these patients presented with hemiplegia upon admission, which persisted postoperatively despite interventional recanalization of the affected cerebral vessel before surgery. One of our patients was extubated after surgery but developed cerebral embolism on his first postoperative day. He underwent a cerebral thrombectomy yet developed hemiplegia. Postoperative arm weakness disappeared in both affected patients. It is worth mentioning that one patient received a TEVAR intervention because of true lumen collapse. medicina-59-01007-t001_Table 1Table 1Preoperative patient characteristics (n = 15).Age (years)58.2 ± 11.1Female gender n (%)6 (40)BMI29.9 ± 5.7Arterial hypertension n (%)12 (80)Diabetes mellitus n (%)2 (13.3)COPD n (%)2 (13.3)Peripheral arterial disease n (%)0 (0)Dialysis n (%)0 (0)History of stroke n (%)4 (26.7)Ejection fraction [%]60 ± 0.0Coronary disease n (%)1 (6.7)Aortic valve regurgitation n (%)11 (73.3)Grade 1 n (%)2 (13.3)Grade 2 n (%)3 (20)Grade 3 n (%)6 (40)Preoperative intubation n (%)0 (0)Conscious n (%)15 (100)Pericardial effusion n (%)4 (26.7)Pericardial tamponade n (%)3 (20)Emergency n (%)13 (86.7)Aortic dissection type—DeBakey 1 n (%)12 (80)Aortic dissection type—DeBakey 2 n (%)2 (13.3)Aortic dissection type—DeBakey 3 n (%)1 (6.7)Dissection of the brachiocephalic trunk n (%)13 (86.7)Dissection of the left carotid artery n (%)10 (66.7)Cerebral ischemia n (%)3 (20)Dissection of the coronary artery n (%)7 (46.7)Coronary ischemia n (%)2 (13.3)Visceral ischemia n (%)0 (0)Dissection of the femoral arteries n (%)2 (13.3)Leg ischemia n (%)1 (6.7)The values are given as the mean ± standard deviation or as the absolute numbers and % in parentheses. BMI: body mass index; COPD: chronic obstructive pulmonary disease.medicina-59-01007-t002_Table 2Table 2Intraoperative data (n = 15).Carotid-stenting before surgery n (%)3 (20)Unilateral cerebral perfusion n (%)11 (73.3)Bilateral cerebral perfusion n (%)4 (26.7)Balloon occlusion of the descending aorta n (%)15 (100)Selective cardiac perfusion n (%)12 (80)Bentall procedure n (%)7 (46.7)Aortic valve reconstruction n (%)1 (6.7)Aortic arch replacement n (%)15 (100)Reconstruction of the aortic root and ST junction n (%)7 (46.7)E-vita prosthesis n (%)14 (93.3)AMDS prosthesis n (%)1 (6.7)Intraoperative angiography n (%)3 (20)Intraoperative carotid-stenting n (%)2 (13.3)Body temperature [°C]28.9 ± 2.3; 30 (27–31)Aortic cross-clamp time [min]91.6 ± 26.9; 90 (72–114)CPB time [min]237.0 ± 55.6; 229 (205–259)Lower body circulatory arrest time [min]8.1 ± 4.2; 8 (5–11)Duration of surgery [min]397 ± 121.9; 387 (292–446)The values are given as the mean ± standard deviation and median (IQR), if applicable, or as absolute numbers and % in parentheses. ST junction: sinutubular junction; AMDS: ascyrus medical dissection stent hybrid prosthesis; CPB: cardiopulmonary bypass; IQR: interquartile range.


## 4. Discussion

In this initial series of 15 unselected patients with type A aortic dissection, our newly applied perfusion concept proved to be technically feasible. The mean circulatory arrest time was 8 min, and surgery was performed at moderate hypothermia (mean lowest body temperature = 29 °C).

Searching for alternatives to deep hypothermic circulatory arrest should be one of the main goals when it comes to modern open aortic arch surgery. In our opinion, our modified perfusion technique further simplifies the procedure and makes it more reproducible [2].

The following maneuvers are responsible for this simplification and have a significant impact on circulatory arrest time during open aortic arch surgery:Perfusion of the heart after completion of aortic root surgeryBalloon occlusion of the descending aorta and perfusion of the lower bodySurgery in moderate hypothermiaProximalization of the distal anastomotic line from zone III to zone 0 and extra-anatomic reconstruction of the perfusion of the head vessels

Ad 1. The heart perfusion technique has been described earlier [1]. It not only shortens the cardiac ischemia time but also allows for an easy check of the suture lines under pressure.

Ad 2. In FET cases, the balloon, inflated in the stented part of the hybrid prosthesis, serves as an endo clamp, allowing for selective perfusion of the lower body with the ascending aorta still open. It results in a significantly shorter circulatory arrest time. In the case of isolated arch replacement without an aneurysm or dissection of the proximal descending aorta, the procedure may even be performed without circulatory arrest at all.

In addition, the guide wire in the true lumen of the aorta helps to prevent malpositioning of the hybrid prosthesis into the false lumen and hereby makes the procedure safer [3].

Ad 3. Clotting is considerably influenced by temperature. In contrast to deep hypothermia, moderate hypothermia preserves a more physiological coagulation function and reduces bleeding [4,5,6,7].

Ad 4. Proximalization of the distal anastomosis from zone III to zone 0 increases the safety of the operation and is an important simplification of the procedure, concerning both ease of construction of the anastomosis as well as bleeding control. Renouncing reimplantation of the left subclavian artery means further simplification in suitable cases. The artery offspring deep in the left hemithorax is often associated with technically demanding anastomosis, carrying an enhanced risk of bleeding, thus increasing CPB and operation time. Ligation of the left subclavian artery has also been reported earlier [8].

Last but not least, interdisciplinary collaboration with our colleagues from the radiology and the vascular surgery departments is crucial to diagnose and find answers to malperfusion syndrome at an early stage.

We would like to mention the limitations of our study. First of all, our study describes a new strategy in complex aortic surgery but is limited to a series of 15 initial cases. Another limitation is the lack of a control group. With larger numbers, we may be able to compare the technique to a patient collective in a matched study; however, the performance of a randomized study does not seem to be a realistic concept. In any case, further investigations will be necessary to fully explore the potential benefits of this promising perfusion strategy.

## 5. Conclusions

In conclusion, we present our modified perfusion approach in complex aortic arch surgery, which allows for considerably reducing circulatory arrest duration and surgery at temperature levels that come closer to the temperature levels at which we perform routine cardiac surgery in fields such as coronary bypass grafting and valve reconstruction or replacement. Future research on larger populations will have to show whether the modifications that considerably simplify surgery can be translated into a clinical benefit for the affected patients.

## Figures and Tables

**Figure 1 medicina-59-01007-f001:**
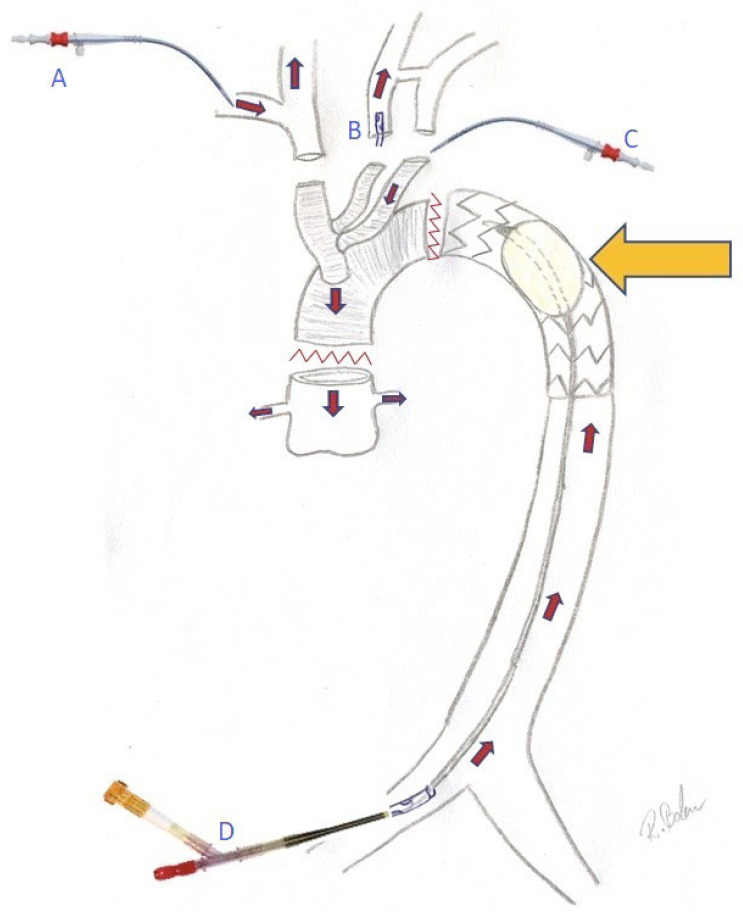
Perfusion strategy after deployment of the frozen elephant trunk. The yellow arrow indicates the inflated occlusion balloon placed in the stented part of the frozen elephant trunk. A cannula in the right axillary artery (A) provides initial whole-body perfusion as well as cerebral perfusion as soon as the brachiocephalic trunk has been clamped. A second cannula in the left carotid artery (B) may be used if necessary. Heart perfusion is provided through a cannula in one of the branches of the prosthesis (C). The ThruPort cannula is placed into one of the femoral arteries (D) and will perfuse the lower part of the body after the inflation of the occlusion balloon (yellow arrow).

**Table 3 medicina-59-01007-t003:** Postoperative patient data (n = 15).

Cardiac ischemic event n (%)	0 (0)
Pacemaker n (%)	0 (0)
Respiration time [min]	114.5 ± 189.7; 72 (10–105)
Reintubation n (%)	1 (6.7)
Pneumonia n (%)	3 (20)
Cerebral ischemic event (including preoperative ischemia) n (%)	7 (46.7)
Hemiplegia n (%)	5 (33.3)
Arm weakness n (%)	2 (13.3)
Cerebral thrombectomy n (%)	1 (6.7)
TEVAR n (%)	1 (6.7)
Permanent dialysis n (%)	2 (13.3)
Sternal wound infection n (%)	0 (0)
ICU stay (days)	18.3 ± 13.7; 9 (8–31)
Hospital stay (days)	23.8 ± 11.7; 20 (15–33)
30-day mortality n (%)	0 (0)

The values are given as the mean ± standard deviation and median (IQR), if applicable, or as absolute numbers and % in parentheses. ICU: intensive care unit; IQR: interquartile range; TEVAR: thoracic endovascular aortic repair.

## Data Availability

Not applicable.

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
