# Peer review of "A Strategy for Minimizing Circulatory Arrest Duration in Complex Aortic Arch Procedures"

_medicina, 2023, doi:10.3390/medicina59061007_

Round 1
Reviewer 1 Report
Article is well written and deals with an interesting topic. Avoiding deep hypotermia and reducing ischemic arrest during distal anastomosis in surgery for TAAD is always crucial and deserves investigation. Methods are accurate. I would recommend to include a figure / picture summarizing the surgical approach. References should be expanded as many recent papers have faced this topic.
English language should be revised by a native speaker to improve fluency.
Author Response
We would like to thank the reviewer for his constructive comments.
We have added Figure 1. to the manuskript, herein we present an overview of our concept.
Four additional references were added.
Reviewer 2 Report
Balan et al. describe a special perfusion strategy for shortening the (lower body) circulatory arrest time during complex aortic arch surgery after acute aortic dissection. By usage of a frozen elephant trunk with a simultaneous endoclamp technique, it was possible to reduce circulatory arrest time to 8.1 mins at moderate hypothermia.
Overall, the authors present an innovative and promising surgical technique for treatment of acute aortic dissection. The manuscript is well written and the clinical data seems to be promising, however, some revisions should be made:
- The authors present a circulatory arrest time of 8.1 mins and a good clinical outcome of their patient collective. However, a control cohort would have been interesting to directly see the clinical benefit of the described technique. Is it possible to add such data?
- Please list specific in- and exclusion criteria (contraindications) for the usage of the presented technique if there have been any.
- More references needed. In general, 4 references are a very low number. Furthermore, specific statements should be supported by a supporting references (i.e. l. 39: “…not perfused for 30-60 min,…”).
- It would be interesting if the authors could compare (benefits / drawbacks) their technique against other surgical strategies in the discussion section.
- Only a minor thing, but please format decimal numbers uniformly (i.e. l. 20 vs. l. 22) and use punctuation marks.
Author Response
- We eould like to thank the reviewer for his constructive comments.
- The comparisson of the patients operated with our novel strategy against a controll group would be subject of a further investigation in our institution
- We have not defined any inclusion or exclusion criteria, of course each of the presented surgical steps can have its own contraindications. Our aim is to follow the same strategy whenever possible, the subjects of this study, were 15 consecutive, non selected patients operated by 2 surgeons in our institution
- Format of the decimal numbers was corrected in l.20-22
- Another four references were added
- The benefits of this technique were described in the discussion section (Ad 1,2,3,4)
Reviewer 3 Report
1. I am not a native English speaker, however, it seems the presentation should be corrected in terms of the style of the language.
2. The concept of “potential re-entries” is not defined in the article, therefore, it requires clarification. This pathology was visualized according to CT, ultrasound or assumed
3. Whу was perioperative carotid intervention necessary in 5 patients (33%)? Was it possible to do this during one operation?
4. I consider it incorrect to include in the analysis a patient who was implanted with AMDS prosthesis. I propose to exclude it from the analysis.
5. What do the authors attribute such a high percentage of neurological complications to? What is more, the article does not specify the protocol of cerebral perfusion.
6. Also given the short time of circulatory arrest of the lower body, the need for 13.3% of cases of permanent dialysis is not discussed. Is it related to acute kidney injury before surgery or during surgery?
7. Pilot study implies a prospective design. In this case, the design is retrospective, so this study cannot be called a pilot study.
8. The list of references does not take into account a number of other relevant works.
Author Response
We would like to thank the reviewer for his valuable comments.
1. We have added some changes to the style and language of the manuskirpt
2. The concept of "potential re-entries" was not defined in the article, therefore it was removed from our manuskript
3. As described in text, all of these patients were operated in a hybrid operating room. We added l101-102 to the our manuskript: Eventual malperfusion syndromes were addressed at the beginning of the surgical procedure
4. Regardless of the used prosthesis (FET, ET, AMDS) the operating technique remained the same.
5. As described in text, three of the seven patients with postoperative neurological complications presented with hemi-plegia upon admission, wich persisted postoperatively. One patient developed cerebral embolism on the first postop day and underwent cerebral thrombectomy.
6. Post-operative dialysis is a common complication of the patients with aortic dissection because of the malperfusion of the kidneys. Further investigations are necessary to asess the long term renal function ot these patients.
7. This is not a pilot study. Our study design was described between the l 50-60.
The word "pilot" was replaced with "initial" in the discussion.
8. Four aditional refrences were included.
Round 2
Reviewer 2 Report
Thank you for submission of the revised manuscript. I also want to mention, that the manuscript benefits from the addition of the nice drawing.
Unfortunately, I cannot see relevant changes in the manuscript and some of my comments were only addressed in the cover letter with the re-submission - Typically, the reviewers comments should be considered and reflected in the revised manuscript.
So, I still see the need for clarification of the following two comments:
- If no control was established initially and the authors are not able to retrospectively add a (ideally propensity-matched) control, please add this as a limitation to the discussion section.
- I can see that benefits of the new techniques have been discussed. However, there is still no discussion (1st) on possible drawbacks and (2nd) against other techniques / publications. The discussion section of a scientific paper should always cover both, the positive (benefits) as well as the negative (drawbacks or potential problems) aspects of a procedure. Please adjust the discussion section accordingly.
Author Response
We would like to thank the reviewer for his constructive comments. We have adjusted the discussion section according to the suggestions of the reviewer and also expanded the figure legend.